# Gene Prioritization through Consensus Strategy, Enrichment Methodologies Analysis, and Networking for Osteosarcoma Pathogenesis

**DOI:** 10.3390/ijms21031053

**Published:** 2020-02-05

**Authors:** Alejandro Cabrera-Andrade, Andrés López-Cortés, Gabriela Jaramillo-Koupermann, César Paz-y-Miño, Yunierkis Pérez-Castillo, Cristian R. Munteanu, Humbert González-Díaz, Alejandro Pazos, Eduardo Tejera

**Affiliations:** 1Grupo de Bio-Quimioinformática, Universidad de Las Américas, Quito 170125, Ecuador; yunierkis.perez@udla.edu.ec; 2Carrera de Enfermería, Facultad de Ciencias de la Salud, Universidad de Las Américas, Quito 170125, Ecuador; 3RNASA-IMEDIR, Computer Sciences Faculty, University of A Coruna, 15071 A Coruña, Spain; aalc84@gmail.com (A.L.-C.); c.munteanu@udc.es (C.R.M.); apazos@udc.es (A.P.); 4Centro de Investigación Genética y Genómica, Facultad de Ciencias de la Salud Eugenio Espejo, Universidad UTE, Quito 170129, Ecuador; cesar.pazymino@ute.edu.ec; 5Laboratorio de Biología Molecular, Subproceso de Anatomía Patológica, Hospital de Especialidades Eugenio Espejo, Quito 170403, Ecuador; gaby_jaramillok@yahoo.com; 6Escuela de Ciencias Físicas y Matemáticas, Universidad de Las Américas, Quito 170125, Ecuador; 7Biomedical Research Institute of A Coruña (INIBIC), University Hospital Complex of A Coruña (CHUAC), 15006 A Coruña, Spain; 8Centro de Investigación en Tecnologías de la Información y las Comunicaciones (CITIC), Campus de Elviña s/n, 15071 A Coruña, Spain; 9Department of Organic Chemistry II, University of the Basque Country UPV/EHU, 48940 Leioa, Spain; 10IKERBASQUE, Basque Foundation for Science, 48011 Bilbao, Spain; humberto.gonzalezdiaz@ehu.es; 11Facultad de Ingeniería y Ciencias Agropecuarias, Universidad de Las Américas, Quito 170125, Ecuador

**Keywords:** gene prioritization, osteosarcoma, communality analysis, pathogenesis, early recognition

## Abstract

Osteosarcoma is the most common subtype of primary bone cancer, affecting mostly adolescents. In recent years, several studies have focused on elucidating the molecular mechanisms of this sarcoma; however, its molecular etiology has still not been determined with precision. Therefore, we applied a consensus strategy with the use of several bioinformatics tools to prioritize genes involved in its pathogenesis. Subsequently, we assessed the physical interactions of the previously selected genes and applied a communality analysis to this protein–protein interaction network. The consensus strategy prioritized a total list of 553 genes. Our enrichment analysis validates several studies that describe the signaling pathways PI3K/AKT and MAPK/ERK as pathogenic. The gene ontology described TP53 as a principal signal transducer that chiefly mediates processes associated with cell cycle and DNA damage response It is interesting to note that the communality analysis clusters several members involved in metastasis events, such as *MMP2* and *MMP9*, and genes associated with DNA repair complexes, like *ATM*, *ATR*, *CHEK1*, and *RAD51*. In this study, we have identified well-known pathogenic genes for osteosarcoma and prioritized genes that need to be further explored.

## 1. Introduction

In recent years, high-throughput technologies have focused on studying the molecular etiology of osteosarcoma (OS) worldwide [1,2,3,4,5]. Valuable information has been gained about whole genetic groups that describe cellular and molecular changes in OS [6,7]. Despite this, there has not been an agreement about specific driver genes for OS etiology, nor have new biomarkers been proposed to be used as therapeutic targets.

OS tumors are characterized by being heterogeneous and showing high rates of somatic structural variations. Their heterogeneity is closely related to their high rates of mutations, which are comparable to breast tumors and leukemia [8,9,10]. Moreover, cytogenetic abnormalities in OS tumors, including chromosomal segment loss, rearrangement, and amplification with karyotypic complexity in the absence of recurrent clonal translocations, have been described [11,12]. This acute chromosomal instability and widespread deregulation in cell signaling pathways could be the main limitations for the description of specific gene drivers associated with OS. It is therefore necessary to develop an integrative study focused on the biology of systems described for this tumor.

The use of prioritization strategies, through computational tools that use multiple heterogeneous data sources, allows for the improvement in gene detection related to complex traits or specific clinical phenotypes [13,14]. In addition, applying the functional enrichment analysis has proven to be a very efficient approach in gene prioritization because it describes important metabolic interactions that aid in explaining the pathogenesis of a given disease [15,16]. Thus, we used several bioinformatics tools in order to prioritize genes that describe oncological signaling pathways for OS and also applied a consensus strategy with the aim to specify and postulate new pathogenic mechanisms that explain the onset and development of this sarcoma. In Figure 1, we summarize the general workflow to prioritize genes associated with the pathogenesis of OS.

## 2. Results

### 2.1. Consensus Prioritization

We chose nine bioinformatics methods that fulfilled two main criteria: full availability in web service platform and only requiring the disease name (or OMIN code, 259,500 for OS) for gene prioritization. In total, the combination of all methodologies resulted in 15,809 genes.

The validation strategy for gene prioritization was performed from the identification of specific genes involved in the OS pathogenesis. For this, we took into consideration pathogenic OS genes defined by a literature review of two types of studies: meta-analysis, based on publications and case reports for OS patients (named as G1 genes), and gene description in animal models and OS cell lines (named as G2 genes). Thereby, we identified 75 pathogenic OS genes from the available literature, of which 47 were classified as G1 and 41 as G2 (Appendix A).

The number of pathogenic genes detected by the nine prioritization tools was lower than our consensus strategy (Table 1). By comparing the number of pathogenic genes detected by all methodologies, our consensus list identifies the highest percentage of those defined as G1 and G2. Specifically, in the top 1% of our consensus method (the first 158 positions), 60% of pathogenic genes (45 of 75) were detected, followed by Genie (35.29%) and Phenolizer (30.14%) methodologies. Furthermore, in the top 20%, the consensus method remains the best at detecting pathogenic genes (88%), followed by Genie, Phenolizer, and SNPs3D with percentages of 80.88%, 72.60%, and 71.88%, respectively.

On the other hand, the mean ranking of the pathogenic genes detected in the top 1% of the list is 49.3 (Table 2), which means that 45 G1–G2 genes are located in the top 50 positions. This mean is higher than that calculated for the other prioritization methodologies, given that the number of pathogenic genes detected is greater. However, it is interesting to note that the number of genes and the ranking average are similar, which indicates that the majority of these pathogenic genes are found in the top positions.

This initial prioritization generated an initial amount of 15,809 genes, so a rational cut-off was applied. The maximum variation between Ii and the gene ranking was 0.7609, corresponding with a ranking value of 553. Therefore, this cut-off reduces a list of 15,809 members to a consensus of 553 genes (Appendix A), which corresponds to 3.5% of the total. The rate of pathogenic detection of the consensus was 87.2% for G1 (41 out of 47), 80.5% for G2 (33 out of 41), and 81.3% for G1 and G2 (61 out of 75), higher than the other methods in the top 5% onwards.

### 2.2. Enrichment Analysis of OS Related Genes and the Protein–Protein Interaction Network

A gene ontology (GO) analysis and pathway enrichment analysis was applied in order to describe biological functions from the consensus genes by using the David Bioinformatics Resource [17,18]. The GO analysis of these 553 consensus genes resulted in 263 terms related to biological processes (Appendix A), adjusted to an FDR *p*-value < 0.01. Using Revigo [19] and only considering terms with a frequency lower than 0.01%, we narrowed our list down to 92 (Appendix A). Some of these specific biological processes are listed in Table 3.

Likewise, the enriched metabolic pathways considered in KEGG and Reactome databases are shown in Appendix A. A partial list of the prioritized metabolic pathways with an FDR *p* < 0.01 is presented in Table 4.

The enriched biological processes of the 553 genes describe terms associated with positive DNA replication, cellular proliferation, and apoptotic events, in which TP53 is one of the most relevant signal transducers. In addition, more specific sarcoma-related terms are listed, such as smooth muscle cell and fibroblast proliferation, osteoblast differentiation and development, and positive regulation of mesenchymal cell proliferation.

The pathway enrichment analysis showed pathways in cancer and the cell cycle in general. The enrichment from the KEGG database showed widely described signaling pathways in cancer in the top positions, for instance, FOXO, PI3K/AKT, TP53, MAPK, neurotrophin, and cell cycle. Moreover, the Reactome database lists events mainly related to cell cycle regulation such as cyclin D-associated events in G1, G0 and early G1; Cyclin A: Cdk2-associated events at S phase entry, and Cyclin A/B1 associated events during G2/M transition.

### 2.3. Protein–Protein Interaction Analysis

We evaluated the physical interactions of the members of the consensus list by including the protein interactions described for *Homo sapiens* from the STRING database [20]. The protein–protein interaction (PPI) generated an osteosarcoma–PPI network (OS–PPI) of 505 nodes from the 553 consensus genes (91.3%). The node degrees of the 58 pathogenic genes (named as G1 and G2) detected in this network were higher than the non-pathogenic ones (39.05 and 19.25, respectively), showing statistical differences when applying the non-parametric Mann–Whitney *U*-test (*p* < 0.001). Therefore, a higher node degree given by this interaction signifies a greater probability of association with pathogenesis within the prioritized genes.

### 2.4. Communality Analysis and Weight of Enriched Pathway

The communality analysis was carried out using the clique percolation method. The clustering data through the communality analysis was obtained with Cfinder [21], which defined “k-cliques” based on the interaction degree of each node from the OS–PPI network and the extent to which different communities overlapped in said network. The clique percolation method allowed us to detect 14 k-cliques and 86 possible communities with a composition of between 17 and 465 genes. The early minimum in $S^{k}$ variation with respect to k-parameters (Figure 2) revealed that *k* = 8 and *k* = 9 have similar gene distributions within communities ($S^{k}$ index 0.719 and 0.609, respectively). Both k-cliques are suitable for further analysis; however, we chose *k* = 9 because it had a better *Mean_rank* (218.89) than *k* = 8 (243.95). Moreover, *k* = 9 is composed of 13 communities and 245 genes (44.3% of the 553 OS genes).

In order to weigh the metabolic pathways obtained in the enrichment analysis, we ranked these terms within each k-clique by means of a pathway enrichment analysis. The pathway enrichment analysis of genes in the 13 communities for *k* = 9 (Appendix A) is consistent with the results obtained in the enrichment analysis (Table 4). As shown in Table 5, P53, cell cycle, and FOXO continue to hold the top positions and ErbB, TGFB and VEGF improved their statistical significance within this k-clique.

To be more selective about which communities would be most relevant within these 13, we used a clustering analysis. The K-means clustering analysis revealed four main community groups (Figure 3). Cluster 1 (Communities 4, 9 and 13) had the highest average values of ${\mathrm{ConsenScore}}_{i}$, ${\mathrm{Degree}}_{i}$ and ${\mathrm{PathScore}}_{m}$, followed by Cluster 2 (Communities 5, 8 and 10) with regards to ${\mathrm{ConsenScore}}_{i}$ and ${\mathrm{Degree}}_{i}$. Therefore, these six communities were chosen for further analysis.

Communities 4, 5, 8, 9, 10, and 13 have groups from 9 to 13 genes and, in total, contain 47 prioritized genes. The genetic distribution among the communities is almost specific and only Communities 4 and 9 present a high similarity (77%) regarding gene composition (Table 6). Only *TP53* is shared in five of the six communities, which denotes its centrality in this prioritization.

Genes in Communities 8, 10, and 13 are highly relevant for the signaling pathways PI3K/AKT and ERBB/MAPK (*PIK3CA*, *PTK2*, *HRAS*, *KRAS*, *SCH1*, *AKT*). In Community 13, the matrix metalloproteases *MMP2* and *MMP2* are prioritized, which together with *FGF2*, reflects processes related to cell migration. Since AKT is a central protein in cellular signaling, several downstream effectors are described in Communities 5 and 8. The genes *ARID1A*, *SMARCE1* and *SMARCB1*, specific to Community 5, are mainly associated with chromatin remodeling.

Given the close metabolic relationship between Communities 5, 8, 10, and 13, it is not surprising that *JUN*, *NFKB1*, *VEGFA*, *TGFB1*, *CREBBP,* and *RELA* are shared among them. However, Communities 4 and 9 are isolated from the rest of the clusters and only have *TP53* in common. The genetic composition of both communities is specific to one biological process: DNA repair. *ATM*, *CHEK1*, *ATR*, *BRCA1*, *BRCA2*, *RAD51*, *BLM,* and *MLH1* belong to DNA repair complexes associated with cellular response to DNA damage stimuli, DNA repair, and double-strand break repair via homologous recombination. Altogether, the genetic distribution of these communities is in accordance with the GO analysis obtained from our consensus list (Table 3).

The 47 genes grouped into the six communities defined above represent the most important prioritized members within this study, so we developed a sub-network based on these results (OS–comms network). The centrality index calculated in this sub-network was significantly correlated with the node degree (${\mathrm{Degree}}_{i}$) of the same genes in the original OS–PPI network (*r* = 0.317, *p* = 0.03).

### 2.5. Gene Validation

As a validation strategy, we compare our consensus list with the DRIVE project (deep RNAi interrogation of visibility effects in cancer) [22] and with the cancer-focused protein–protein interaction network (OncoPPI) [23] data. The data generated by the DRIVE project described 83.5% of our 553 consensus genes (Appendix A). Of these 461 genes, 20 were determined as essential, 70 as active and 371 as inert. On the other hand, the OncoPPI network recognized 92 of our prioritized genes (16.6%) and its centrality index showed a significant correlation with the same gene in our OS–PPI network (*r* = 0.445, *p* < 0.001) (Appendix A).

As shown in Figure 4A, both DRIVE and OncoPPI genes are present in the OS–comms network. From the DRIVE analysis, *BRCA1* and *RAD51* were identified as essential and *ATR*, *CDK2*, *CDK4*, *CHEK1*, *SMARCB1*, *SMARCE1*, *RELA*, *AKT1*, *MYC*, *HRAS* as active. On the other hand, 17 OncoPPI genes (36.18% of 47 in OS–comms network) were present in this network. Upon correlating the centrality indices between the OncoPPI network and the OS-comms network, we obtained a statistical correlation (*r* = 0.512, *p* = 0.036).

We can notice that in Figure 4B, several prioritized genes are actually transcription factors (TFs). Because of this, we chose to perform a second prioritization focused only on TFs and without using PPI networks. The PPI could bias toward physical interactions and reduce the relevance of regulatory mechanism, as presented in TFs. We identified 125 TFs from the initial 553 genes already prioritized. The ${\mathrm{TFscore}}_{i}$ was evaluated for all TFs (Appendix A). The top 20 more relevant TFs are *TP53*, *E2F1*, *JUN*, *RUNX2*, *FLI1*, *YY1*, *HIF1A*, *MYC*, *TP63*, *ESR1*, *WT1*, *E2F4*, *ATF2*, *NFKB1*, *AR*, *SP1*, *STAT1*, *ERG*, *CEBPB*, *TFAP2A*.

From the 125 TFs, 4% were identified as essential and 9.1% as active when compared to DRIVE genes. Additionally, 19 TFs were present in the OncoPPI network. Regarding community analysis, 27.6% were TFs and were mainly present within Communities 5, 8 and 13 (Figure 4B).

## 3. Discussion

As shown in Table 1, the detection rate of our consensus prioritization strategy was higher than all the bioinformatics tools employed in this analysis. Moreover, the mean rank of the pathogenic genes detected in the top 1% of the list was 49.3. Table 2 indicates that, on average, the 45 G1–G2 genes were located in the top 50 positions. These results confirm that this methodology does indeed improve the detection and prioritization of pathogenic genes, as had been previously described in other pathologies [24,25].

As a first approach, the prioritization strategy resulted in a consensus list of 553 genes and the 10 top-ranked genes were *TP53*, *RB1*, *CHEK2*, *RUNX2*, *E2F1*, *MDM2*, *CDKN1A*, *JUN*, *CCNA2* and *CDKN2A*. *TP53*, *RB1*, *CHEK2,* and *MDM2* were ranked in 1st, 2nd, 3rd, and 6th positions, respectively, and also the arrangement of the pathogenic genes in this list shows a distribution in the top positions. So far, the gene ranking along this prioritization reflects a proper gene weighting based mainly on this consensus strategy. These genes had been previously described in OS pathogenesis. Early studies focused on the molecular biology of OS were carried out on individuals with familial syndromes, which predisposed them to this tumor. Germline inactivation of *RB1* and *TP53* were initially described in patients with hereditary retinoblastoma and Li–Fraumeni syndrome, respectively [26,27], and subsequently in sporadic sarcomas [28,29]. Given that these two suppressors are central proteins in controlling the cell cycle, later studies briefly described many others that interacted with them. Mouse double minute 2 (*MDM2*), for example, is a protein that binds to RB1 and inactivates TP53 [30]. Its amplification is an event that occurs in primary OS (3–25%) and it is overexpressed in metastases and recurrences [31,32]. CHEK2 is another protein that is part of a DNA damage checkpoint, works as a stabilizer of TP53, and shows a 7% frequency of mutations in OS patients [33,34].

The biological processes derived from the GO analysis of the 553 genes describe TP53 as a principal signal transducer that mediates processes associated with cell cycle, DNA damage response, DNA replication and intrinsic/extrinsic apoptotic signaling regulation. Additionally, more specific biological processes were described, for instance, fibroblast proliferation, osteoblast differentiation and development, and mesenchymal cell proliferation and transition. In accordance with our results, previous studies have identified similar biological processes related to OS, where the following are considered OS-associated terms: cell cycle regulation (mainly mediated by RB1 and TP53), osteoblast differentiation (mediated by RUNX2), DNA damage, stress response, epigenetic processes, mitosis, cell motility functions, and members involved in OS cell proliferation (weighting NFKB signaling, NFKBIE, and RELA members) [3,35,36,37]. Taken together, these processes suggest that the consensus list is evidence of the genes associated with osteogenesis, cell differentiation, and transition to bone cell types. In addition, the terms derived from the pathway enrichment analysis (Table 4) are in accordance with these biological processes.

The information used by STRING allowed us to define the degree of physical interaction of the consensus list members and calculate their centrality index. This centrality index was used as a variable to evidence the contribution rate of the pathogenic genes to a common biological purpose. Thus, the greater the centrality for a node within the OS–PPI network, the greater the probability of its contributing to pathogenesis. This association was validated by analyzing the genes defined as pathogenic (G1–G2), in which significant differences were observed in comparison with the rest of the consensus genes (*p* < 0.0001). The centrality index calculated from the 503 nodes included in the protein–protein interaction network determined *TP53* as the most central node, followed by *AKT1*, *MYC*, *JUN*, *EP300*, *CREBBP*, *CCND1*, *CDKN1A*, *STAT3,* and *RB1*. Furthermore, this degree allowed for the definition of more specific clusters and prioritization of gene communities associated with OS pathogenesis. Thus, k-9 was determined as the clique with the best gene distribution among all the resulting communities ($S^{k}$ index 0.719) and Communities 4, 5, 8, 9, 10, and 13 as the most important groups of genes within our study.

The pathway enrichment analysis for the *k* = 9-clique results in, almost in its entirety, the same terms obtained from the initial consensus list. This confirms that the gene filtered through the communality analysis comprised almost the same biological processes. Considering the ${\mathrm{PathScore}}_{m}$ (Table 5), the P53 signaling pathway and cell cycle are in the top positions. FOXO also increases its significance in this enrichment analysis. In different cancer types, PI3K/AKT, Ras-MEK-ERK, IKK, and AMPK are the most important signaling pathways interacting with FOXO [38]. The gain of function of P13K and RAS, or PTEN disruption, are oncogenic events that promote a loss of function in the Forkhead Box transcription factors (*FOXO*) [39]. Interestingly, loss of its expression promotes impaired osteogenic differentiation, suggesting that *FOXO1* is involved in osteoblastogenesis and osteoclastogenesis [40,41,42]. Moreover, FOXO members have an important role in cell fate decision, via triggering the expression of death receptor ligands like FASLG, TNF apoptosis ligand, and some BCL-2 family members (*BCL2L1*, *BNIP3*, *BCL2L11*) [43,44,45,46]. FOXO expression in OS tumors is low or even lacking altogether, leading to tumor progression and cell cycle arrest [47]. The fact that *FOXO* enhances its weight within our enrichment analysis demonstrates its importance as a signaling pathway in the pathogenesis of OS. Furthermore, the close relationship between the FOXO signaling pathway and cell cycle, events of osteoclast differentiation and apoptosis via the TNF signaling pathway, is evidenced in the pathway enrichment analysis applied to the consensus list and the *k* = 9 clique.

Our consensus strategy seeks to specify a group of genes that describe the molecular etiology of OS. In this sense, the use of all the strategies previously described prioritizes to a great extent the 47 genes arranged in Communities 4, 5, 8, 9, 10, and 13. From these six communities, *BRCA1*, *AKT1*, *ATR*, *CDK4*, *HRAS*, *MYC*, *PIK3CA*, *RELA*, *STAT3* are genes validated by DRIVE and Onco–PPI (19.1%), *RAD51*, *CDK2*, *CHEK1*, *SMARCB1*, *SMARCE1* are validated only by DRIVE (10.6%), and *ATM*, *CDH1*, *EGFR*, *EP300*, *ERBB2*, *JUN*, *NFKB1*, *SHC1*, *TP53*, *SP1* by Onco–PPI (21.3%). The sub-network generated from these communities (OS–comms network) reflects closely interrelated genes at the cellular interaction level (Figure 4B) and also groups of genes immersed in important oncological processes. Tamborero et al. [48], from exome sequencing data of 3205 tumors in the Cancer Genome Atlas (TCGA) research network, proposed 291 high-confidence cancer driver genes acting on 12 different cancer types. Although in this study, data from samples of bone tumors were not taken into account, their results showed the members of the PI3K signaling pathway as central onco-drivers, ATR-BRCA1 as regulatory nodes of repair processes associated with TP53, CHEK1 and AKT as the main regulators of cell cycle in function of CDK1A, and CDK1B and activators for downstream pathways such as FOXO. This experimental data support our findings, where *PIK3CA*, *AKT1*, *PTEN*, *HRAS* and *SHC1* were nodes highly connected within our OS–comms network. Nodes that connect to Communities 10 and 13 describe genes representative of our weighted tumorigenic pathways, PI3K/AKT and MAPK/ERK.

The findings reported here suggest that PI3K/AKT and MAPK/ERK are the main signaling pathways deregulated for OS. Several reports have shown that these pathways are responsible for controlling cellular processes related to proliferation, growth, differentiation, and apoptosis [49,50]. In fact, the Ras/Raf/MEK/ERK pathway is hyperactivated in 30% of human cancers [51] and nearly 67% of OS shows aberrant ERK activation [52]. The extra cellular-signal-regulated kinases (ERK) promote cell proliferation, cell survival, and metastasis, particularly by its upstream activation from EGFR and the G protein-coupled receptor Ras [53]. The presence of *SHC1*, *EGFR*, *HRAS*, *PIK3CA*, *ERBB2* within Community 10 support this scenario for OS. In addition, the high connectivity of the matrix metalloproteases, *MMP2* and *MMP9,* in Community 13 suggests a metastasis event in the function of these signaling pathways.

Although the invasion of tumor cells is a general characteristic in carcinogenesis, metastasis to the lung is one of the main characteristics in patients with OS and one of the major causes of mortality [54,55], so this event is a hallmark for this sarcoma. Pathogenic events, including cellular detachment from primary tumors, matrix remodeling and invasion from tumor cells, angiogenesis, vascular dissemination, and proliferation at new sites, are involved in tumor metastasis [56,57]. Upstream regulators of MAP/ERK signaling such as *IL6*, *VEGFA,* and *FGFR1* demonstrate an important role in this process [58,59,60,61,62] and are prioritized in our results. In addition, Community 13 shows the *MMP2* and *MMP9* genes with a high centrality index. A high expression of MMP9 was observed in metastatic OS samples [63,64], leading to speculation that this metalloproteinase can promote cell migration and invasion in OS by degradation components of the extracellular matrix. This evidence suggests that *MMP2* and *MMP9,* together with upstream regulators of MAP/ERK signaling such as *IL6*, *FGF2*, *VEGFA*, *EGFR* and *ERBB2*, are pathogenic nodes dependent on the centrality of PI3K/AKT and MAPK/ERK. This finding could be related to aspects of invasiveness and prognosis, mainly in tumors that present deregulation in these two signaling pathways.

In addition to evidencing the previous findings, Communities 4, 5 and 9 include genes widely described in processes of homologous recombination (HR), base excision repair, and chromatin modification. Cells DNA damage response principally involves maintaining chromosome integrity and genome stability and implies recognition of DNA lesions, followed by an activation of the checkpoints in the cell cycle that promotes cellular signaling cascades related to DNA repair. While the ATM-CHEK2 pathway is responsible for the initiation of cellular responses to double-strand breaks [65,66], ATR-CHEK1 responds to DNA replication stress by means of the phosphorylation of several substrates in response to agents such as UV and X-ray among others [67]. *ATM*, *ATR,* and *CHEK1* show a high centrality index in the OS–comms network, interacting in addition to *BRCA1* and *RAD51*, described as essential genes, and with the cyclin-dependent kinases, *CDK2* and *CDK4*, described as active ones according to the DRIVE validation. Checkpoint activation by ATM mainly controls G1/S, whereas ATM and ATR contribute to establishing and maintaining the S and G2/M checkpoints [68]. Either by activation of ATR-CHEK1 or ATM-CHEK2, DNA damage signaling promotes inhibition of CDK activity and therefore the activation of G1/S, intra-S, and G2/M checkpoints [69]. Consequently, it is likely that such nodes associated with DNA repair, such as *ATM*, *ATR*, *CHEK1*, *BLM*, *RAD51* and *MLH1* (as shown in our pathway enrichment analysis), together with those previously described (*BRCA1* and *BRCA2*) from exome sequencing [70], have important implications regarding the deregulation of the cell cycle evidenced in OS.

While it is true that the nodes described for Communities 4 and 9 are mainly related to repair and cell cycle control events, the HR repair complex is involved in a hallmark event for sarcomas, such as alternative telomere maintenance (ALT). Several molecular details of this mechanism still remain unknown; however, two distinctive telomere phenotypes are described for ALT in human telomerase-negative cells (ALT cells) such as long and heterogeneous telomere DNA and promyelocytic leukemia (PML) body [71], together forming the ALT-associated promyelocytic leukemia body (APB). The PML body is a nuclear made up of proteins which form amongst the chromatin and is related to a wide range of cellular processes including tumors formation, cellular senescence, and DNA repair [72,73]. Numerous lines of evidence strongly suggest that the ALT pathway is dependent on HR since several proteins involved in DNA double-strand break (DSB) are localized at APBs [74,75,76,77]. It is significant that proteins localized at APBs, such as PML, DNA helicases of the RecQ family (*BLM*, *WRN* and *RECQL4*), *RAD51* and *RAD52* (a member of the MNR complex), rank highly in our prioritization. In this sense, the members belonging to HR complexes are described as repair complexes in response to DNA damage. They are relevant to the pathogenesis of the OS, not only as factors immersed in cell cycle control, as previously discussed, but also because they are involved in processes of chromosome stability given by telomere maintenance [78,79,80,81]. Consistent with the literature, where bone tumors are termed as highly heterogeneous, highly mutable, and genetically unstable, members described in Communities 4 and 9 (*TP53*, *ATM*, *ATR*, *CHEK1*, *BLM*, *BRCA1*, *BRCA2*, *RAD51*, *MLH1*, *CDK2*, *CDK4*) explain many of these key features within OS, and can also be associated with important clinical characteristics such as tumor aggressiveness, metastasis, and poor survival.

The use of the GTRD database allowed us to define the frequency of interaction of each TF with the 553 prioritized genes. It is worth noticing that more than half of the prioritized factors (103, 82.4%) interacted with more than half of all genes at the same time. This suggests that more than 80% of the genes defined as TFs actively regulated the genes associated with the pathogenesis of OS. The weight given to each one of these TFs through interaction analysis places the following genes on the top positions: *TP53*, *E2F1*, *JUN*, *RUNX2*, *FLI1*, *YY1*, *HIF1A*, *MYC*, *TP63*, *ESR1*, *WT1*, *E2F4*, *ATF2*, *NFKB1*, *AR*, *SP1*, *STAT1*, *ERG*, *CEBPB*, and *TFAP2A*. When compared to total prioritization, genes *E2F1*, *JUN*, *RUNX2*, *FLI1*, *YY1*, *HIF1A*, *MYC*, *TP63*, *ESR1*, *WT1*, *E2F4*, *ATF2,* and *NFKB1* significantly improved their ranking. During the G1 phase of the cell cycle, RB1 suppresses the function of the E2F1, E2F2, and E2F3 TFs. Sequential hypo phosphorylation of RB1 by cyclin-dependent kinases, CDK4 and CDK6, and CDK2, led in the release of E2F and transcription of genes necessary for cell cycle progression, including cyclins A, D, and E [82]. The improved score of these TFs suggests that these deregulation events in the cell cycle are basal within the pathogenesis of the OS. Although this scenario is common for all types of cancer, a deeper study of the *E2F1* and *E2F4* genes, and depending on those prioritized in Communities 4 and 9 along with *TP53*, would be necessary to define driver proteins in OS tumors.

We identified *TP53*, *JUN*, *MYC*, *ATF2*, *NFKB1*, *SP1*, *CEBPB*, *STAT3*, *KLF4*, *RELA*, *NR3C1*, *CEBPD,* and *PPARG* as TFs (13 or the 47 nodes) in the OS–comms network. The new ranking calculated for each of them improved significantly when compared to the ranking of all OS genes (Appendix A). This suggests that their degree of regulation within this network is very significant and shows evidence of its importance as regulatory proteins within each prioritized cluster.

TFs were grouped over Communities 5, 8 and 13. With *TP53* as the central node, *JUN* and *MYC* are key factors in the pathogenesis of the OS that regulate signaling associated with the pathogenic pathways PI3K/AKT and MAPK/ERK. Furthermore, the prioritization of TFs evidenced *NFKB1* as a central node in these three communities. Nuclear factor-kappa B1 (*NF**κB1*) is a pleiotropic transcription factor that contributes to tumorigenesis in many types of cancer. It works as a key regulator of a variety of genes implicated in many biological events including cell survival, differentiation, apoptosis, and autophagy [83]. When observing the OS–comms network, the high degree interaction of *AKT* with respect to *JUN*-*MYC*, *TGFB1*, *NFKB1*, and *BCL2* suggests this cluster as an important group in the OS pathogenesis. GO terms listed in Table 3 are in accordance with these findings since its activation promotes many types of downstream signaling including osteoblast differentiation via TGFB1 and NFK1 or apoptosis via BCL2 [84,85].

In conclusion, the use of a consensus strategy proved to be efficient when specifying a broad list of genes obtained from several bioinformatics prioritization tools. In addition, the combination of these strategies with a network enrichment analysis allowed us to show not only real interactions between specific genes but also to define internal interactions that explained cellular events associated with OS pathogenesis. Our results validate several studies that describe the signaling pathways PI3K/AKT and MAPK/ERK as oncological for OS. Nevertheless, given its centrality at the cellular signaling level, its deregulation can influence downstream specific pathways, such as FOXO, and promote tumorigenic scenarios like osteoblast undifferentiation via *TGFB1* and *NFK1*, apoptosis via *BCL2,* and migration and metastasis mediated mainly by *MMP2* and *MMP9*.

What is more, the gene composition of Communities 4 and 9, and more specifically to their *ATM*, *ATR*, *CHEK1,* and *RAD51* genes, suggest that the HR repair complex is an important group of genes within the pathogenesis of the OS. Its deregulation can influence tumorigenic events characteristic of this sarcoma as generalized disruption in the cell cycle and ALT mechanisms. Hence, it is necessary to experimentally validate these results, taking into account not only the patient’s age group but also genetic factors that can influence the molecular behavior of these bone tumors, such as racial and ethnic factors. It should also be interesting to study genetic variants of the transcription factors identified and their relationship with possible disease prevalence.

## 4. Materials and Methods

### 4.1. Prioritization Methods and Consensus Strategy

The bioinformatics methods used in this study were for gene-disease prioritization Biograph [86], Cipher [87], DisGeNET [88], Génie [89], GLAD4U [90], Guildify [91], Phenolizer [92], PolySearch [93], and SNPs3D [94]. We chose these nine bioinformatics methods because (1) they are fully available on web service platforms and (2) they only required the disease name (or OMIN code, 259,500 for OS) for gene prioritization. With the disease name and/or the OMIM code, a list of prioritized genes was obtained from each method. Each of these methods follows several different strategies for gene prioritization, and as a final output, they also provide different scores for each gene.

The strategy applied to integrate the gene scores obtained in each independent method is similar to that previously described [24,25]. Thus, we normalized each gene (denoted as *i*) from the ranked list obtained from each method (denoted as *j*) (${\mathrm{GeneN}}_{i,j}$ which means, the normalized score of the gene “*i*” in the method “*j*”). The final score by gene (${\mathrm{ConsenScore}}_{i}$) was considered as the average normalized score and the number of methods which predict the gene (denoted as *n_i_*) are
(1)$${\mathrm{ConsenScore}}_{i}\;=\sqrt{\left(\frac{\left(n_{i}-1\right)}{9-1}\right)\left(\frac{1}{j}\sum_{j}{\mathrm{GeneN}}_{i,j}\right)}$$

This equation refers to the geometric mean between the average score of each gene derived from each method, and the normalized score according to the number of methods that predict the association of the gene and the disease. This consensus approach will lead to a big final list of genes ranked according to the ${\mathrm{ConsenScore}}_{i}$. In order to reduce this list, we needed to follow some rational strategy.

From a manual observation and curation of the scientific literature, we create a list of genes that are highly probable to be involved in OS pathogenesis (Appendix A). For this, we took into consideration pathogenic OS genes defined by a literature review of two types of studies: meta-analysis, based on publications and case reports for OS patients (named as G1 genes), and gene description in animal models and OS cell lines (named as G2 genes). Thereby, we identified 75 pathogenic OS genes from the available literature, of which 47 were classified as G1 and 41 as G2. These manually curated genes were used for (1) validation of the prioritized genes (and networks) and (2) to reduce the initial list of consensus genes.

The pathogenic OS genes (defined as G1 and G2 in Appendix A) were used to calculate $I_{i}\;=\frac{{\mathrm{TP}}_{i}}{{\mathrm{FP}}_{i}+1}{\mathrm{ConsenScore}}_{i}$, where TP and FP are the true and false positive values (up to the ranking value of the gene *i*), respectively. According to that which has been previously described [24,25], the maximum value of $I_{i}$ can be taken as the maximum compromise between the TP and FP rates compensated with the ranking index of each gene. The ranking (“i”), at which “I” _”i” is maximal, will represent a rational cut-off for the consensus list.

We applied another prioritization methodology to demonstrate the degree of interaction of all transcription factors in our consensus genes. We used the “The Human Transcription Factors” database [95] to identify the TFs from the 553 initially prioritized genes. The second prioritization of only TFs was carried out considering the ${\mathrm{ConsenScore}}_{i}$ (Equation (1)) for each TFs and the interaction degree of each TF using the information described in the GTRD (Gene Transcription Regulation Database) [96]. This database contains experimental information from ChIP-seq experiments of TF binding sites. Data were systematically collected and uniformly processed using a special workflow (pipeline) for a BioUML platform (http://www.biouml.org). By inspecting all the target genes described for *Homo sapiens*, we downloaded the information of all the genes defined as TF and all the genes or +/- 5000bp that contain a GTRD meta cluster for this TF.

Thus, the ${\mathrm{TFscore}}_{i}$ was calculated as
(2)$${\mathrm{TFscore}}_{i\;}=\sqrt{\left(\frac{\left(t_{i}-1\right)}{553-1}\right)ConsenScore_{i}}$$
where *t_i_* is the number of genes that are regulated by the transcription factor “*i*”. The general conception is that a transcription factor will be more relevant if it has a higher value in the consensus score and regulate many of the prioritized genes.

### 4.2. Protein–Protein Interaction Network Analysis

The protein interactions of the members of the consensus list were revised from the STRING database, only taking into consideration interactions with a confidence cut-off of 0.9. With this information, we generated a OS–PPI network with zero node addition. Network visualization and analysis were carried out through the Cytoscape software [97].

### 4.3. Communality and Pathway Enrichment Analysis

The communality analysis on OS–PPI was carried out using the clique percolation method with Cfinder [21]. The communality analysis provides a topology description of the network including the location of highly connected sub-graphs (cliques) and/or overlapping modules that usually correspond with relevant biological information. The selection of the value “k-cliques” (*k* = 1,2,3…n) will affect the number of community and also the number of genes in each community. In general, higher values of k-cliques imply few communities while lower values lead to many communities. In the OS–PPI network, both extremes (too small or too high k-cliques values) result in an unbalanced distribution of the genes across communities. This means some of the communities will have a big amount of genes while others will have a very small number.

In order to determine the best k-clique in the communality analysis, we used the index *“S”* [24,25]: $S^{k\;}=\frac{\;\left|\mathrm{mean}\left(N_{g}^{k}\right){-\mathrm{median}(N}_{g}^{k})\right|}{N_{c}^{k}}$, where $N_{g}^{k}$ and $N_{c}^{k}$ are the number of genes in each community and the number of communities for a defined k-clique cut-off value. If the distribution of genes across communities is close to a Gaussian distribution, or constant, $S^{k}$ will tend toward 0. Once k is defined in k-clique, a number of communities will be identified.

Additionally, we applied the partitional algorithm K-means in order to define our best communities within a k-clique. The variables used for the clustering were the means of ${\mathrm{ConsenScore}}_{i}$, ${\mathrm{Degree}}_{i}$, and ${\mathrm{PathScore}}_{m}$ for each community within the k-clique. The ${\mathrm{Degree}}_{i}$ variable refers to the node’s degree centrality index calculated for each gene from the OS–PPI network and the ${\mathrm{PathScore}}_{m}$ is outlined below. From communities selected in this clustering, we created a sub-network to visualize the interactions of all the members of the chosen communities.

For the pathway enrichment analysis, we used a *PathRankScore_m_*, *PathGeneScore_m_*, and *PathScore_m_* as described previously [24]: (1) Each community “*k*” was weighted as $W_{k}\;=\sum {\mathrm{ConsenScore}}_{i}^{k}{/N}_{k}$, where ${\mathrm{ConsenScore}}_{i}^{k}$ is the ${\mathrm{ConsenScore}}_{i}$ of the gene “*i*” in the community “*k*” and $N_{k}$ is the number of communities; (2) Each pathway “*m*” was weighted as ${\mathrm{PathRankScore}}_{m}\;=\sum W_{k}^{m}{/N}_{k}^{m}$, where $W_{k}^{m}$ is the weight ${(W}_{k}$) of each community connected with the pathway “*m*” and $N_{k}^{m}$ is the number of communities connected with the pathway “*m*”, and (3) A second weight to the pathway “*m*”, ${\mathrm{PathGeneScore}}_{m}$, considered all the genes included in each pathway:${\;\mathrm{PathGeneScore}}_{m}\;=\sqrt{{\mathrm{ConsenScore}}_{i}^{m}\frac{n_{m}}{N_{m}}}$, where *N_m_* is the total number of genes in the pathway “*m*”, while *n_m_* is the number of those genes that are also found in the protein–protein interaction network. The average of the ${\mathrm{ConsenScore}}_{i}$ of all genes presents in the pathway “*m*” is $\left\langle {\mathrm{ConsenScore}}_{i}^{m}\right\rangle$. The geometrical mean between ${\mathrm{PathGeneScore}}_{m}$ and the normalized ${\mathrm{PathRankScore}}_{m}$ refers to the final score associated with the pathway “*m*” (${\mathrm{PathScore}}_{m}$).

### 4.4. Gene Validation with the OncoPPi OS Network and the DRIVE Project

Besides the genes in the G1 and G2 groups, we also used the information in the DRIVE project. It is a project that describes a comprehensive mapping of cancer genes obtained from a larger-scale gene knockdown experiment in 398 cancer cell lines. We filtrated the results of eight cell lines, all of which had pathological annotations related to bone cancer (A673, SAOS2, SJSA1, SKES1, SKNMC, SW1353, TC71, and U2OS). Subsequently, all essential genes that showed a *Sensitivity Value* of ≤ −3 in >50% of the chosen cell lines, active genes that showed values of ≤ −3 in 1–49%, and inert ones showed values of ≤ −3 for 0% of cancer cells [22] were compared with our results.

Additionally, from Onco–PPI Portal (http://oncoppi.emory.edu/) [23], a cancer-focused protein–protein interaction network was generated by only considering the interactions described for bone tumor types (labeled as OncoPPI). This network was comprised of 171 genes and 442 interactions. The Spearman correlation of ${\mathrm{Degree}}_{i}$ between the OncoPPI, OS–PPI, and the sub-network from the identified communities were calculated.

## Figures and Tables

**Figure 1 ijms-21-01053-f001:**
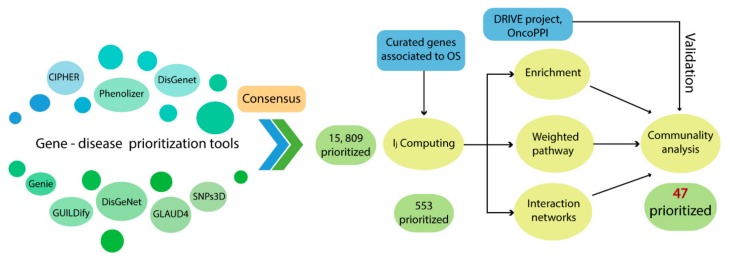
General workflow to gene prioritization.

**Figure 2 ijms-21-01053-f002:**
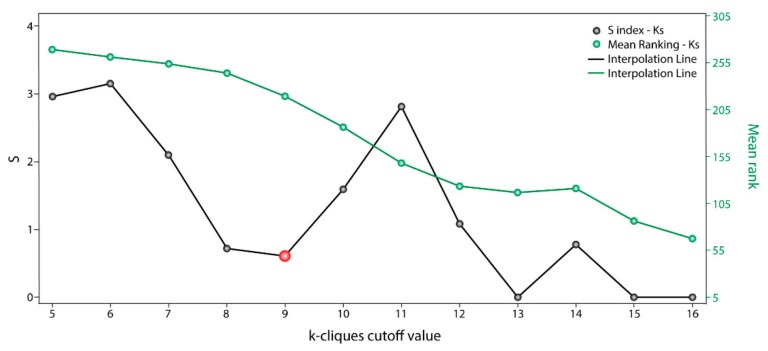
$S^{k}$ scoring with respect to each k-clique cutoff value. Communality analysis by clique percolation method. Values of $S^{k}$ (black points) and mean rankings (green points) with respect to each k-clique cutoff value.

**Figure 3 ijms-21-01053-f003:**
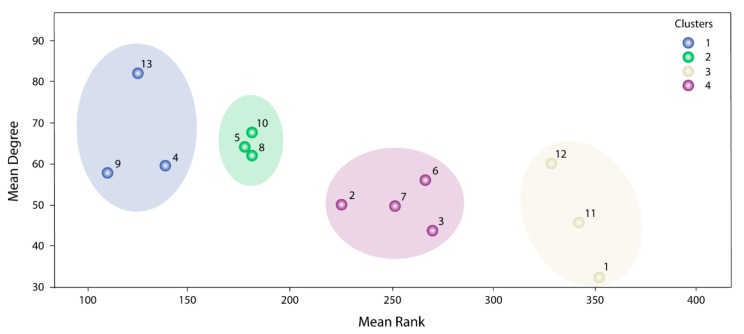
Clustering analysis for the *k* = 9 communities. Blue circles represent Cluster 1, purple circles Cluster 2, yellow circles Cluster 3 and purple circles represent Cluster 3.

**Figure 4 ijms-21-01053-f004:**
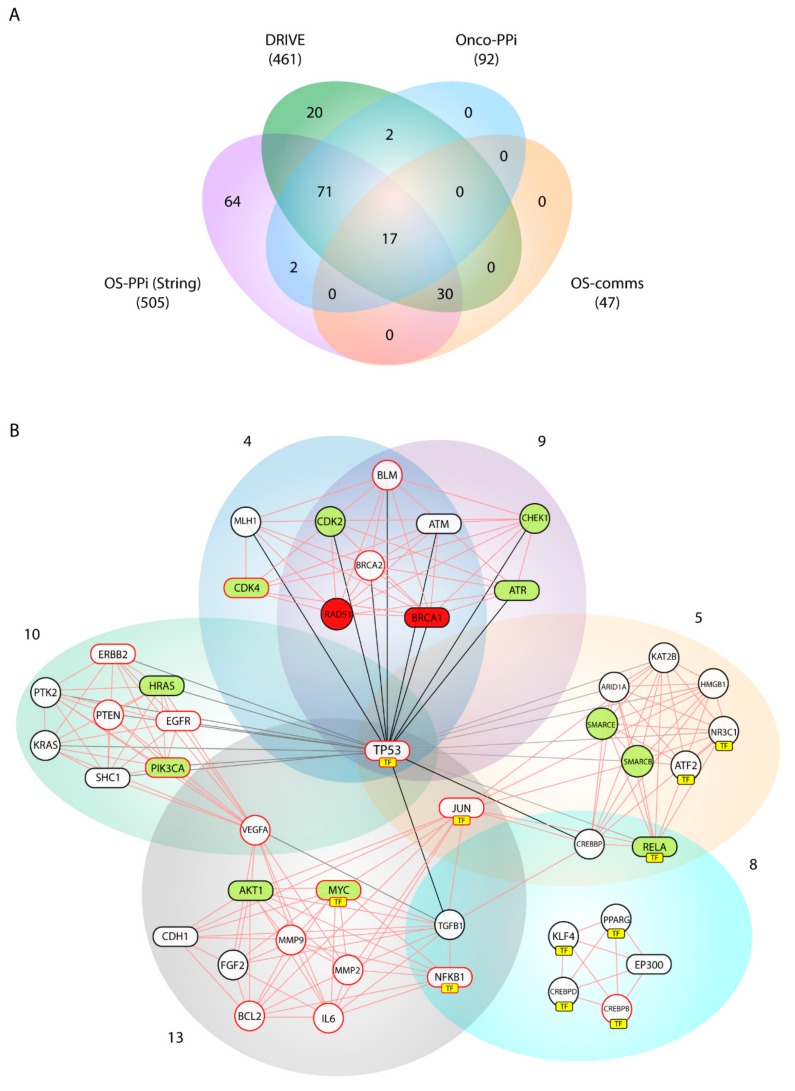
Gene validation and network analysis of the *k* = 9-clique. (**A**) Comparison of prioritized genes from STRING (OS-PPI), DRIVE Project, OncoPPi network, and Cfinder analysis; (**B**) Network analysis from Communities 9, 13, 4, 5 8, and 10 (OS–comms network). Red and green painted nodes are defined as essential and active genes, respectively, based on the results from the DRIVE project. Nodes enclosed in rectangles belong to the analyzed OncoPPI network. Nodes with red borders are members of G1 and G2. Yellow boxes (TF) point to nodes identified as transcription factors.

**Table 1 ijms-21-01053-t001:** Identification (in %) of pathogenic genes in each osteosarcoma (OS) approach.

Methods	1%			5%			10%			20%		
	G1	G2	G1-2	G1	G2	G1-2	G1	G2	G1-2	G1	G2	G1-2
BioGraph	0	0	0	0	18.2	12.5	40	45.5	37.5	60	54.6	50
CIPHER	7.7	6.7	8.7	7.7	6.7	8.7	23.1	20	17.4	30.8	26.7	26.1
DisGeNET	9.5	16.7	10.8	21.4	30.6	21.5	42.9	58.3	46.2	57.1	77.8	64.6
Genie	37.8	36.1	35.3	62.2	61.1	57.4	75.6	69.4	70.6	86.7	75	80.9
GLAD4U	0	0	3.6	19.1	33.3	25	42.9	50	46.4	57.1	66.7	64.3
GUILDify	10.9	7.5	8.2	13	7.5	9.6	21.7	17.5	19.2	34.8	25	30.1
Phenolizer	33.3	36.6	30.1	57.8	61	53.4	62.2	61	56.2	77.8	75.6	72.6
PolySearch	0	0	0	11.1	14.3	7.1	11.1	28.6	14.3	11.1	28.6	14.3
SNPs3D	10	10.5	6.3	10	42.1	25	40	57.9	50	75	73.7	71.9
Consensus	66	61	60	87.2	80.5	81.3	89.4	82.9	84	93.6	85.4	88

**Table 2 ijms-21-01053-t002:** Rank of pathogenic genes in each OS approach.

Methods	1%			5%			10%			20%		
	G1	G2	G1-2	G1	G2	G1-2	G1	G2	G1-2	G1	G2	G1-2
BioGraph	-	-	-	-	3.5	3.5	7	6	6.3	9.5	7.3	8
CIPHER	2	7	4.5	2	7	4.5	41.3	43	32.8	58	59	57.7
DisGeNET	5.3	4.2	4.7	12.1	10	11.1	23.9	23.6	25.2	31.6	31.4	33.7
Genie	17	14.6	16.5	44	41.6	42.6	88.2	75	91.3	148.5	113.2	151.9
GLAD4U	-	1	1	4	4.2	4	8.6	6.6	8.2	13.3	10.2	13
GUILDify	15.8	8.3	16.7	42.6	8.3	43.3	366.5	536.4	491.2	873.8	973.9	972.1
Phenolizer	44.3	28	36.4	150.4	120.9	148	200.9	120.9	182.5	477.5	429.2	513.2
PolySearch	-	-	-	2	2	2	2	2.5	2.5	2	2.5	2.5
SNPs3D	1.5	1.5	1.5	1.5	6.4	4	17.8	10.9	14.4	27.1	16.2	21.6
Consensus	54.5	41.6	49.3	126.1	108.2	128	152.9	131.2	157.7	241.4	174.7	239.3

**Table 3 ijms-21-01053-t003:** Some biological processes by enrichment analysis in OS consensus genes.

BP ID	Name	Frequency	Log10 p-Value (FDR)
GO:1901796	regulation of signal transduction by p53 class mediator	0.01%	−22.8416
GO:0006977	DNA damage response, signal transduction by p53 class mediator resulting in cell cycle arrest	0.00%	−20.1656
GO:0048661	positive regulation of smooth muscle cell proliferation	0.01%	−16.5544
GO:0048146	positive regulation of fibroblast proliferation	0.01%	−16.5031
GO:0045740	positive regulation of DNA replication	0.01%	−15.1965
GO:1902895	positive regulation of pri-miRNA transcription from RNA polymerase II promoter	0.00%	−14.983
GO:0043525	positive regulation of neuron apoptotic process	0.01%	−14.9393
GO:0071260	cellular response to mechanical stimulus	0.01%	−13.3507
GO:0032355	response to estradiol	0.01%	−11.7258
GO:0045669	positive regulation of osteoblast differentiation	0.01%	−11.5058
GO:0060395	SMAD protein signal transduction	0.01%	−11.1904
GO:0042771	intrinsic apoptotic signaling pathway in response to DNA damage by p53 class mediator	0.01%	−10.8356
GO:0097192	extrinsic apoptotic signaling pathway in absence of ligand	0.01%	−10.0846
GO:0035019	somatic stem cell population maintenance	0.01%	−9.6162
GO:0010332	response to gamma radiation	0.01%	−9.4056
GO:0002053	positive regulation of mesenchymal cell proliferation	0.01%	−9.2628
GO:0002076	osteoblast development	0.00%	−9.1046
GO:0048538	thymus development	0.01%	−8.2907
GO:0048010	vascular endothelial growth factor receptor signaling pathway	0.01%	−7.6946
GO:0010718	positive regulation of epithelial to mesenchymal transition	0.01%	−7.6126

**Table 4 ijms-21-01053-t004:** Pathways enrichment analysis using KEGG and Reactome databases in OS consensus genes.

Pathway ID	Pathway Name	% Genes	FDR
KEGG Database			
hsa05200	Pathways in cancer	26.22	1.33 × 10^−8^
hsa04110	Cell cycle	11.93	3.96 × 10^−45^
hsa04068	FoxO signaling pathway	10.85	4.50 × 10^−35^
hsa04151	PI3K-Akt signaling pathway	15.55	1.98 × 10^−29^
hsa05206	MicroRNAs in cancer	14.1	3.07 × 10^−29^
hsa04115	p53 signaling pathway	7.23	2.38 × 10^−29^
hsa05205	Proteoglycans in cancer	11.57	1.63 × 10^−27^
hsa04210	Apoptosis	6.69	6.13 × 10^−26^
hsa04668	TNF signaling pathway	8.32	2.89 × 10^−25^
hsa04510	Focal adhesion	10.85	4.08 × 10^−23^
hsa04380	Osteoclast differentiation	8.68	1.21 × 10^−22^
hsa04010	MAPK signaling pathway	11.75	8.86 × 10^−22^
hsa04722	Neurotrophin signaling pathway	7.78	1.68 × 10^−19^
hsa04012	ErbB signaling pathway	6.69	2.67 × 10^−19^
hsa04917	Prolactin signaling pathway	5.79	3.57 × 10^−17^
hsa04914	Progesterone-mediated oocyte maturation	6.33	3.70 × 10^−17^
hsa04014	Ras signaling pathway	9.76	3.87 × 10^−16^
hsa04550	Signaling pathways regulating pluripotency of stem cells	7.41	7.26 × 10^−15^
hsa04919	Thyroid hormone signaling pathway	6.69	9.79 × 10^−15^
hsa04350	TGF-beta signaling pathway	5.79	1.28 × 10^−14^
**REACTOME Database**			
R-HSA-69231	Cyclin D associated events in G1	4.7	5.00 × 10^−21^
R-HSA-1538133	G0 and Early G1	3.44	1.13 × 10^−15^
R-HSA-69656	Cyclin A:Cdk2-associated events at S phase entry	2.35	1.10 × 10
R-HSA-69273	Cyclin A/B1 associated events during G2/M transition	2.71	1.42 × 10
R-HSA-2173796	SMAD2/SMAD3:SMAD4 heterotrimer regulates transcription	3.07	4.22 × 10
R-HSA-1257604	PIP3 activates AKT signaling	4.34	5.16 × 10^−9^
R-HSA-5674400	Constitutive Signaling by AKT1 E17K in cancer	2.53	4.01 × 10^−8^
R-HSA-2219530	Constitutive Signaling by Aberrant PI3K in cancer	3.62	6.77 × 10^−8^
R-HSA-69202	Cyclin E associated events during G1/S transition	1.99	9.93 × 10^−8^
R-HSA-1912408	Pre-NOTCH Transcription and Translation	2.53	4.36 × 10^−7^

**Table 5 ijms-21-01053-t005:** Pathways enrichment analysis of *k* = 9 communities and their associated weights.

Pathway Name	PathScore_m_	Community
p53 signaling pathway	0.603	2, 4, 9, 10
Cell cycle	0.595	2, 4, 7, 8, 9, 13
FoxO signaling pathway	0.578	2, 7, 8, 10, 11, 12, 13
Prolactin signaling pathway	0.574	2, 8, 10, 12
ErbB signaling pathway	0.565	2, 10, 11, 12, 13
Central carbon metabolism in cancer	0.564	2, 10, 11, 12, 13
TGF-beta signaling pathway	0.553	2, 6, 7, 8
Pathways in cancer	0.546	2, 3, 4, 5, 6, 7, 8, 9, 10, 11, 12, 13
VEGF signaling pathway	0.536	2, 10, 11, 12
Adherens junction	0.534	2, 3, 6, 7, 8, 10, 11, 12
Proteoglycans in cancer	0.534	2, 10, 11, 12, 13
HIF-1 signaling pathway	0.532	2, 5, 6, 7, 8, 10, 11, 12, 13
Choline metabolism in cancer	0.526	2, 10, 11, 12
Thyroid hormone signaling pathway	0.524	1, 2, 3, 5, 6, 7, 10, 13
TNF signaling pathway	0.523	2, 5, 8, 13
NOD-like receptor signaling pathway	0.522	2, 8, 13
Osteoclast differentiation	0.52	2, 8, 11, 12, 13
Focal adhesion	0.518	2, 10, 11, 12, 13
Progesterone-mediated oocyte maturation	0.518	2
Apoptosis	0.515	2, 4, 5, 8, 9, 10, 13
Neurotrophin signaling pathway	0.515	2, 5, 10, 11, 12, 13
Fc epsilon RI signaling pathway	0.514	2, 10, 11, 12
MicroRNAs in cancer	0.508	2, 4, 8, 9, 10, 12, 13
mTOR signaling pathway	0.504	2, 10
B cell receptor signaling pathway	0.502	2, 5, 8, 10, 11, 12, 13

**Table 6 ijms-21-01053-t006:** Gene distribution in the most relevant communities in *k* = 9-clique.

Comms	Genes	Mean *ConsenScore_i_*	Mean *Degree*	Mean *PathScore_m_*	Pathogenic Genes/Genes
9	*TP53, ATM, BRCA1, CHEK1, CDK2, ATR, BRCA2, RAD51, BLM*	0.802	57.78	0.656	0.333
13	*TP53, JUN, VEGFA, MYC, MMP2, BCL2, MMP9, NFKB1, IL6, FGF2, AKT1, TGFB1, CDH1*	0.776	81.85	0.598	0.692
4	*TP53, CDK4, ATM, BRCA1, CDK2, BRCA2, RAD51, MLH1, BLM*	0.751	59.33	0.656	0.444
5	*TP53, JUN, ATF2, CREBBP, SMARCB1, HMGB1, KAT2B, RELA, ARID1A, NR3C1, SMARCE1*	0.68	64	0.594	0.182
8	*NFKB1, SP1, CREBBP, CEBPB, CEBPD, STAT3, KLF4, EP300, RELA, PPARG, TGFB1*	0.675	62	0.612	0.273
10	*TP53, VEGFA, EGFR, PTK2, ERBB2, SHC1, PTEN, PIK3CA, HRAS, KRAS*	0.673	67.4	0.599	0.6

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
