# Peer review of "Gene Prioritization through Consensus Strategy, Enrichment Methodologies Analysis, and Networking for Osteosarcoma Pathogenesis"

_ijms, 2020, doi:10.3390/ijms21031053_

Round 1

Reviewer 1 Report

The manuscript entitled  "Gene Prioritization Through Consensus Strategy Enrichment Methodologies Analysis and Networking for Osteosarcoma Pathogenesis“ combine consensus prioritization and downstream computational analysis to identify pathogenic genes for osteosarcoma. This is a comprehensive study and reveal many potential important gene for osteosarcoma, but still require more effort to prioritize gene significance. 

1. Driver genes are different for different cancer type, using DRIVE data as gene validation is not appropriate. It's better to include differential expression data and mutation data to further demonstrated selected genes.

2. Most analysis are descriptive and have limit impact on pathogenic genes or disease OS. Could you further prioritize gene significance?

Author Response

Point 1: Driver genes are different for different cancer type, using DRIVE data as gene validation is not appropriate. It's better to include differential expression data and mutation data to further demonstrated selected genes. 

Response 1: We used two strategies for validation: a manual curation of literature leading to the genes reported in G1 and G2 (Supplementary Table S1) and DRIVE. However, from DRIVE we extracted only cell lines related with bone cancer and only in this group we selected the genes for validation. The use of G1 and G2 genes was not properly explained in the previous version of the manuscript. They are also present in Figure 4B but a mistake in the figure caption was not pointing out their presence in this network which represent the final list in our prioritization. Genes obtained from DRIVE were more important in the validation of the network topology (correlations between node centralities between different networks) than in the validation of the final prioritization. We can notice that in Figure 4B there are many genes also present in G1 and G2 (even more than those derived from DRIVE). This relevance should be clearer in the current version of the manuscript after the corrections made (Figure 3 caption and methodology).

We agree that a better validation of final data could be the inclusion of expression or mutation data. However, at the time we started this project few (and consistent) information was available in ArrayExpress of GEO databases. Recently TARGET project from NIH fully released OS data (https://ocg.cancer.gov/programs/target/projects/osteosarcoma). This data could be valuable for validation and actually differentiate possible rolls of the final 47 prioritized genes. However, as you surely realize, this is a challenge that will require a considerable time and analysis surely leading to a second publication.

Point 2: Most analysis are descriptive and have limit impact on pathogenic genes or disease OS. Could you further prioritize gene significance?

Response 2: It is a very important observation. The main goal of the manuscript was to prioritize genes related with OS pathogenesis and we reduced to a group of 47 genes (from initially 15 000 genes). We try to validate the methodology and the final genes using different approaches: curated reported genes, DRIVE project and scientific evidence. However, until this point is not possible to go further in function, mechanism or even to be more confident about the pathogenic role of the final genes without further experimental data.

We started to work on the TARGET data recently released at full which include important amount of reliable experimental information. We are confident that this data could improve our analysis and allow us to move from a descriptive approach toward an explicative and mechanistic based presentation. However, it is an ongoing research.

Reviewer 2 Report

This paper shows an interesting prioritization of genes involved with Osteosarcoma Pathogenesis. The approach used is sound and conclusions are interesting. The authors demonstrated that the results obtained are in line with well assessed databases of consensus genes (like the DRIVE project).

I have appreciated the quality of the presentation along with the clear writing style. The only concern I have is about the input dataset. It is almost unclear how and where the initial set of genes is taken. Perhaps the authors intended that all known genes are to be considered firstly and further filtered with such methodology. This should be clarified in the manuscript in the introduction section otherwise for the readers will be difficult to understand the starting point.

I would also suggest the author to introduce the  communality analysis with a short description: how it works and what it produces. This will make such method more comprehensively described (section 2.4  line 139).

Author Response

Point 1: I have appreciated the quality of the presentation along with the clear writing style. The only concern I have is about the input dataset. It is almost unclear how and where the initial set of genes is taken. Perhaps the authors intended that all known genes are to be considered firstly and further filtered with such methodology. This should be clarified in the manuscript in the introduction section otherwise for the readers will be difficult to understand the starting point. 

Response 1: The author appreciates the reviewer comments. The reviewer is correct that the initial definition of genes was not properly explained in the methods. In the current version of the manuscript we expand the methodology around dataset and we also add a new figure with a clear representation of the methodology.

Point 2: I would also suggest the author to introduce the communality analysis with a short description: how it works and what it produces. This will make such method more comprehensively described (section 2.4 line 139).

Response 2: We extended the description of the communality analysis in the Methodology that was actually lacking in the previous version of the manuscript.

Round 2

Reviewer 1 Report

The manuscript entitle "Gene Prioritization Through Consensus Strategy,Enrichment Methodologies Analysis and Networking for Osteosarcoma Pathogenesis" use prioritization strategies to filter genes involved in Osteosarcoma pathogenesis. In addition, the authors also applied gene enrichment analysis and communality analysis reveal well-known pathogenic genes for osteosarcoma and prioritized genes that need to be further explored.
Genes are also validation by DRIVE project (deep RNAi interrogation of visibility effects in cancer). 

Comments

Communality analysis of protein-protein interaction are mainly used to identify physical connected gene modules. Besides key modules in protein-protein interaction, similar analysis can also used in gene regulator network. Pathways enrichment analysis reveal p53 signaling pathway and Cell cycle pathway. It seems that many genes have regulatory relationship. There are many ChIP-seq and transcription factor target gene databases, it will be useful if manuscript can also prioritize key regulator TFs.

Author Response

Point 1: Communality analysis of protein-protein interaction are mainly used to identify physical connected gene modules. Besides key modules in protein-protein interaction, similar analysis can also used in gene regulator network. Pathways enrichment analysis reveal p53 signaling pathway and Cell cycle pathway. It seems that many genes have regulatory relationship. There are many ChIP-seq and transcription factor target gene databases, it will be useful if manuscript can also prioritize key regulator TFs.

Response 1: We appreciate this observation and included a new prioritization for all the transcription factors identified in our consensus genes. This new strategy is consistent with our previous results, and significantly improves the findings already described.

Thus, we applied another prioritization methodology to demonstrate the degree of interaction of all transcription factors in our consensus genes. We used the “The Human Transcription Factors” database [1] to identify the TFs from the 553 initially prioritized genes. The second prioritization of only TFs was carried out considering the ConsenScorei (Eq.1) for each TFs and the interaction degree of each TF using the information described in the GTRD (Gene Transcription Regulation Database) database [2].
Thus, the TFscorei was calculated as (formula available in the attachment, too) :

TFscorei=√((ti-1)/(553-1))ConsenScorei

Where ti is the number of genes which are regulated by the transcription factor “i”. The general conception is that a transcription factor will be more relevant if it has a higher value in the consensus score and regulate many of the prioritized genes.

The results of the prioritization of TFs are quite similar to that obtained by the prioritization of OS genes, which shows a high rate of detection of pathogenic genes when we use all the prioritization and validation strategies. The use of the GTRD database allowed defining the frequency of interaction of each TF with the 553 prioritized proteins. It is interesting to note more than half of the prioritized factors (103, 82.4%) interact with more than half of all genes at the same time. This suggests that more than 80% of those defined as TFs actively regulate the genes associated with the pathogenesis of OS.

The weighting given to each of these TFs through interaction analysis, places in the top positions the TP53, E2F1, JUN, RUNX2, FLI1, YY1, HIF1A, MYC, TP63, ESR1, WT1, E2F4 , ATF2, NFKB1, AR, SP1, STAT1, ERG, CEBPB, and TFAP2A genes. When compared to total prioritization, the genes E2F1, JUN, RUNX2, FLI1, YY1, HIF1A, MYC, TP63, ESR1, WT1, E2F4, ATF2 and NFKB1 significantly improve their ranking.

All these new analyzes and results have been included in the new version of the manuscript. Supplementary table 2 and figure 4 have been updated. In addition, the discussion of the new prioritized TFs is developed in the Discussion section.

REFERENCES
1. Lambert, S. A.; Jolma, A.; Campitelli, L. F.; Das, P. K.; Yin, Y.; Albu, M.; et al. The Human Transcription Factors. Cell 2018, 175, 598-599.
2. Yevshin, I.; Sharipov, R.; Kolmykov, S.; Kondrakhin, Y.; Kolpakov, F. GTRD: a database on gene transcription regulation-2019 update. Nucleic Acids Res 2019, 47, D100-D105.
